# Deformation Prediction System of Concrete Dam Based on IVM-SCSO-RF

**Shi Zhang** [1,2], **Dongjian Zheng** [1,2,*] **and Yongtao Liu** [1,2]

1   College of Water Conservancy and Hydropower Engineering, Hohai University, Nanjing 210024, China
2   State Key Laboratory of Hydrology-Water Resources and Hydraulic Engineering, Hohai University, Nanjing 210098, China
*   Correspondence: zhengdj@hhu.edu.cn

**Abstract:** Deformation prediction is an important part of concrete dam safety monitoring. In recent years, the random forest (RF) algorithm has attracted more and more attention in the field of dam safety monitoring because of its fast speed and strong generalization ability. However, the performance of RF is easily affected by many factors, such as the drift of measured value in displacement and the inappropriate setting of parameters of RF. To solve the above problems, the indicator variable model (IVM) is used to identify and eliminate the drift of measured values in this paper, and the sand cat swarm optimization (SCSO) is applied to optimize RF for the first time. On the grounds of this, a deformation prediction system of a concrete dam based on the IVM and RF algorithm optimized by SCSO is proposed. The case study shows that IVM can correct the interference of monitoring data accurately, and the maximum error rate is less than 3%; in the aspect of parameter optimization of RF, the results of the SCSO algorithm are obviously better than those of the TAE method and PSO algorithm, and the corresponding OOB error is the minimum; in terms of prediction performance, compared with TAE-RF, PSO-RF, LSTM and SVM, SCSO-RF has higher accuracy and stronger stability, and its SSE and MSE are reduced by at least 91%, MAE and RMSE are reduced by at least 71%, and $R^2$ is very close to 1. The results of study provide a new method for the automatic online evaluation of dam safety performance.

**Keywords:** concrete dam; deformation prediction; sand sat swarm optimization; random forest; indicator variable model

## 1. Introduction

In the 21st century, a new stage of water conservancy construction has been entered, marked by the Xiaowan, Xiluodu, Jinping, Merowe, Yeywa, and other large-scale hydropower stations in the world [1–3]. With the construction of more and more large-scale water conservancy projects, more attention has begun to be paid to the safety of ultra-high dams. As one of the main types of high dams in the world, concrete dams account for more than 60% of the dams with a height of more than 200 m in the world and 56% of the dams with a height of more than 200 m in China [4]. With the continuous breakthrough of design and construction technology [5], more high concrete dams will be built in the future to meet the human demand for water resources. Therefore, the safety of concrete dams plays a crucial role.

Concrete dams are nonlinear dynamic evolution systems affected by multiple factors [6,7]. In the process of long-term service, concrete dams not only bear the effects of various dynamic and static cyclic loads, but also bear various sudden disasters and the long-term combined effects of severe environmental erosion and material performance degradation. These adverse factors greatly increase the risk of concrete dam failures [8]. To effectively prevent and reduce social and economic losses and casualties caused by concrete dam failures, it is necessary to carry out long-term safety monitoring of concrete dams,

identify abnormal dam signs timely and accurately, and implement scientific and reasonable emergency measures to ensure the safe and stable operation of the dam [9,10].

Deformation monitoring is intuitive and reliable, which is the main content of concrete dam safety monitoring at home and abroad. The deformation prediction system of concrete dams built according to the monitoring values of deformation and environment factors can effectively reflect the change law of dam displacement and predict the development trend of deformation, and then scientifically analyze the evolution of its structural properties and evaluate the service state of the dam accordingly [11,12]. At present, the most commonly used deformation prediction method is the statistical model, which is easy to understand and realize, and its accuracy can meet the needs of the project. However, when the influence factors of dam deformation are complex and there are multiple collinearities among variables, its prediction performance is poor [13]. With the development of data mining and artificial intelligence techniques, machine learning algorithms such as support vector machine (SVM), artificial neural network (ANN), long short-term memory (LSTM), adaptive neuro-fuzzy inference system (ANFIS), least-squares support vector machine (LSSVM), and extreme learning machine (ELM) have been widely used in dam deformation prediction [14–20], further improving the accuracy of deformation prediction. However, these machine learning algorithms have shortcomings in practical application; for example, it is difficult to select superparameters of SVM [19], the neural network model has problems of under/overfitting and computational overload [20], and LSTM takes a long training time and cannot process long sequence data.

The random forest (RF) algorithm was proposed by Leo Breiman [21] in 2001. By introducing the idea of ensemble learning, it solves the nonlinear mapping problem between multiple parameters effectively. Because of advantages of RF such as fast training speed, strong generalization ability and no need for cross-validation, it has become an important data-mining method and has been widely used in bioinformatics, ecology, medicine, geomorphology, and other fields [22–25]. In recent years, the RF algorithm has also gradually received attention in the field of dam safety monitoring. Dai et al. [26] studied the method of using the statistical model and RF to predict the deformation of concrete dams, and verified the feasibility of this method through actual monitoring data of the concrete dam. Belmokre et al. [27] applied the RF algorithm to predict dam displacement on the basis of temperature field calculated by the one-dimensional deterministic model, and compared the prediction results with the statistical model and artificial neural network model to verify the prediction performance of the algorithm. Su et al. [28] proposed an improved RF model based on the sliding time window strategy to predict the dam displacement. Gu et al. [29] established an evaluation model of influencing factors of concrete dam performance by using evidence theory and RF algorithm, and verified the mining ability of the model for influencing factors of dam deformation in practical engineering application. It can be seen from previous studies that the parameters of RF are the key factor to determine the prediction accuracy of the algorithm, but the common method is to determine the parameters by the trial-and-error (TAE) method in the parameter range set according to current experience [28]. The TAE method is highly subjective, and it is difficult to achieve satisfactory prediction accuracy in the actual application process. Therefore, how to find the optimal parameters accurately and efficiently is an important problem to be solved in the current application of the RF algorithm.

Meanwhile, the preprocessing of data should be performed very carefully, as important details can be eliminated. All data anomalies should be documented and analyzed properly, namely compared with engineering inferences and dam damage due to aging/cracking, which is especially important for old dams, anomalous weather, geological displacement on the nearby fault, and seismic events [30–32]. Drift of measured value in displacement is a common data anomaly, and the drift caused by reasonable reasons such as engineering reinforcement or maintenance and renovation of the monitoring system belongs to normal interference. However, the RF algorithm is unable to identify and deal with such interfer-

ence in the training process. If the measured value with drift is directly input into RF, it will have a great negative impact on its performance of prediction.

In conclusion, the drift of measured value in displacement and the setting of parameters of RF bring great difficulties to the application of RF in the prediction of concrete dam deformation. These problems greatly limit the performance of RF. Therefore, to reduce the impact of the drift of measured value in automatic monitoring data on the prediction system, the indicator variable model (IVM) is used to preprocess the monitoring data. At the same time, in view of the shortcomings of parameter optimization in the RF algorithm, the sand cat swarm optimization (SCSO) algorithm, which has excellent optimization performance, is used in this study to optimize the RF to obtain the optimal parameter combination for the first time. On the ground of this, a concrete dam deformation prediction system based on the IVM and RF algorithm optimized by SCSO is proposed in this study. The system first identifies and processes the drift of measured value by establishing IVM, and then constructs the deformation prediction model through RF optimized by the SCSO algorithm. Furthermore, the system proposed in this paper is applied to the actual concrete gravity dam, and the results fully verify that the system has excellent capabilities of nonlinear data mining and prediction, which can be used in practical projects.

## 2. Methodology

### 2.1. Indicator Variable Model

Engineering reinforcement or maintenance and renovation of monitoring system often cause interference to the automatic observation system of dams, resulting in a sudden change and large drift of measured values. These abnormal measurements will affect the training and prediction effect of the prediction model greatly. To obtain the measured value that conforms to the actual law and reflects the real change of the effect quantity, an indicator variable model (IVM) based on the adaptive model of analysis of dam automatic observation data proposed by Zheng et al. [33] was constructed to correct the measured value after interference in this study.

To characterize whether the automatic observation system is disturbed in the IVM, it is necessary to introduce indicator variables and find quantitative indicators for them. Here, indicator variables with values of 0 and 1 are used. Let $V$ represent the indicator variable of the system subject to sudden disturbance. When the system is disturbed, $V = 1$; when there is no interference, $V = 0$. For the observation system, the interference is usually transmissible. Therefore, if the system is interfered with when $t = t_i$, the indicator variable can be expressed as

$$V_i = \begin{cases} 1 & t \geq t_i \\ 0 & t < t_i \end{cases} \tag{1}$$

where $t$ is the time; $i$ is the number of times the observation system is interfered, which ranges from 1 to M; $t_i$ is the time of the $i$th interference; and $V_i$ is the indicator variable of the $i$th interference.

Taking the automatic observation data of dam horizontal displacement as an example, the IVM is established. The main factors that affect the horizontal displacement are water pressure, temperature, and time effect. If the observation system is disturbed when $t = t_i$, the all-time IVM can be expressed as

$$\delta = \delta_H + \delta_T + \delta_\theta + \sum_{i=1}^{M} V_i d_i \tag{2}$$

where $\delta$ is the displacement value, $\delta_H$, $\delta_T$, and $\delta_\theta$ are the water pressure component, temperature component, and time effect component of the observation system after the interference is eliminated, and these components are fitted by the statistical model [34]; $d_i$ is the drift caused by the $i$th interference.

Assume that the estimated value of observation value $y$ is $\hat{y}$, and $\hat{y}$ is calculated by Equation (2), then the residual square sum of estimated value $\hat{y}$ and measured value $y$ is

$$Q = \sum_{j=1}^{N} [\hat{y}_j - y_j]^2 = \sum_{j=1}^{N} [\delta_H + \delta_T + \delta_\theta + \sum_{i=1}^{M} V_i d_i - y_j]^2 \tag{3}$$

where $y_j$ and $\hat{y}_j$ represent the $j$th observation value and its estimated value, respectively, and $N$ represents the total number of samples.

According to the principle of least-squares method, it can be obtained that

$$\frac{\partial Q}{\partial a_k} = 0, \ \frac{\partial Q}{\partial b_p} = 0, \ \frac{\partial Q}{\partial c_q} = 0, \ \frac{\partial Q}{\partial d_i} = 0 \tag{4}$$

where $a_k$, $b_p$, and $c_q$ are regression coefficients of water pressure component, temperature component, and time effect component, respectively.

The drift $d_i$ caused by each interference can be calculated by solution (4); it is removed from the measured value in each disturbed period, and then the measured value is corrected.

### 2.2. Sand Cat Swarm Optimization Algorithm

Sand cat swarm optimization (SCSO) algorithm is a new algorithm inspired by nature for solving global optimization problems, proposed by Amir Seyyedabbasi and Farzad Kiani [35] in 2022. This algorithm originates from research on the search and hunting behavior of sand cats. The instinct of sand cats to detect low frequencies below 2 kHz can help them to catch prey at long distance in a short time, and they also have a strong ability to dig prey. Therefore, SCSO algorithm is proposed based on the above two characteristics of sand cats. The algorithm introduces adaptive adjusted auditory sensitivity, randomly initialized search space, and random angle to strengthen the balance between the global search and the local search. It can effectively avoid the local extremum and has higher convergence accuracy and faster speed to solve complex optimization problems. In addition, SCSO algorithm has fewer parameters and is easier to implement than other metaheuristic algorithms.

#### 2.2.1. Initial Population

Unlike the habit of sand cats living independently in nature, SCSO assumes that they are collective. For a $d$ dimension optimization problem, the population of cats is an $n \times d$ array, where $n$ represents the number of cats, which is set according to the actual precision demand and calculation efficiency. The variable value $x_i$ ($i = 1 \sim d$) of each dimension represents the potential solution of the optimization problem and must be located between the upper and lower boundaries. The fitness value of each cat can be obtained by substituting the variable value into the fitness function. After the best individual is evaluated according to the fitness value, other cats move towards it.

#### 2.2.2. Searching the Prey

The search mechanism of sand cats depends on low-frequency perception. To balance the global and local search capabilities of the algorithm better, SCSO introduces adaptive adjusted auditory sensitivity $\overrightarrow{r_G}$. As the iterative process progresses, $\overrightarrow{r_G}$ decreases linearly from 2 to 0 to ensure that cats approach the prey gradually without losing or skipping, which can be expressed as

$$\overrightarrow{r_G} = S_M - \left( \frac{S_M \times iter_c}{iter_{Max}} \right) \tag{5}$$

where $S_M$ is a parameter representing the auditory characteristics of sand cats, and its default value is 2, which can be flexibly adjusted for different optimization problems; $iter_c$ is the current iteration; $iter_{Max}$ is the maximum iterations.

In the search process, the search space is initialized based on the defined boundary randomly, and position updating of each current search agent is based on a random position, so that the search agent can restart the search in the new space. The sensitivity of each cat $\vec{r}$ is defined according to $\vec{r_G}$ to avoid the algorithm falling into the local extremum. Its expression is

$$\vec{r} = \vec{r_G} \times rand(0,1) \tag{6}$$

When the iteration is $t + 1$, the position of each cat will be dynamically updated according to the optimal position $\vec{P_{bc}}$, current position $\vec{P_c}$ and sensitivity $\vec{r}$ after the iteration $t$. The specific update formula is as follows:

$$\vec{P}(t+1) = \vec{r} \cdot \left( \vec{P_{bc}}(t) - rand(0,1) \cdot \vec{P_c}(t) \right) \tag{7}$$

Equation (7) ensures that the new position of each sand cat is between the current position and the position of prey. Therefore, with updating of the position, SCSO algorithm can continuously find new local optimum.

2.2.3. Attacking the Prey

In the stage of hunting, the position update mode of sand cats is different. As shown in Figure 1, assuming that the sensitivity range of the sand cat is a circle, SCSO uses the roulette wheel selection algorithm to select a random angle for each cat to approach the prey. At the same time, using the random angle can further avoid falling into the local optimum. The position of each sand cat at iteration $t + 1$ can be obtained based on its sensitivity, random angle, and the distance from the optimal position after iteration $t$. The specific formula is as follows:

$$\overrightarrow{P_{rnd}} = \left| rand(0,1) \cdot \vec{P_b}(t) - \vec{P_c}(t) \right| \tag{8}$$

$$\vec{P}(t+1) = \vec{P_b}(t) - \vec{r} \cdot \overrightarrow{P_{rnd}} \cdot \cos(\theta) \tag{9}$$

where $\overrightarrow{P_{rnd}}$ is the distance between the optimal position $\vec{P_b}(t)$ and the current position $\vec{P_c}(t)$ at iteration $t$; $\theta$ is a random angle between 0 and 360 degrees.

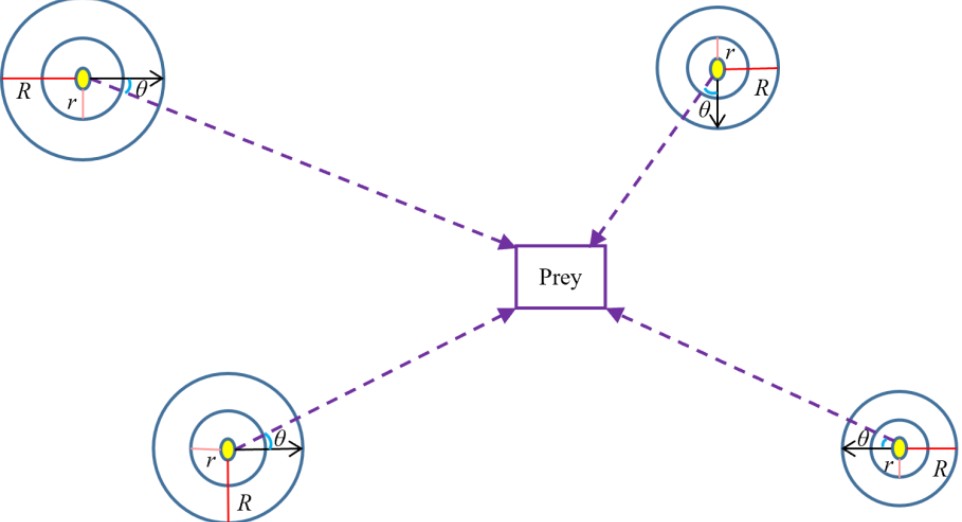

**Figure 1.** Schematic diagram of hunting stage in SCSO.

### 2.2.4. Transition between Search and Hunting Stages

The unique advantage of SCSO is that it can achieve seamless switching between search and hunting through parameter $\vec{R}$, which can be expressed as

$$\vec{R} = 2 \times \vec{r_G} \times rand(0,1) - \vec{r_G} \tag{10}$$

Since $\vec{r_G}$ decreases linearly from 2 to 0 throughout the iteration process, the fluctuation range of parameter $\vec{R}$ is very clear. In the SCSO algorithm, when the absolute value of the parameter $\vec{R}$ is less than or equal to 1, the search agent is forced to enter the hunting stage; otherwise, the search agent is forced to search, which can be expressed as

$$\vec{P}(t+1) = \begin{cases} \vec{P_b}(t) - \vec{r} \cdot \overrightarrow{P_{rnd}} \cdot \cos(\theta) & |R| \le 1 \\ \vec{r} \cdot \left( \overrightarrow{P_{bc}}(t) - rand(0,1) \cdot \vec{P_c}(t) \right) & |R| > 1 \end{cases} \tag{11}$$

The specific flow of SCSO algorithm is shown in Figure 2.

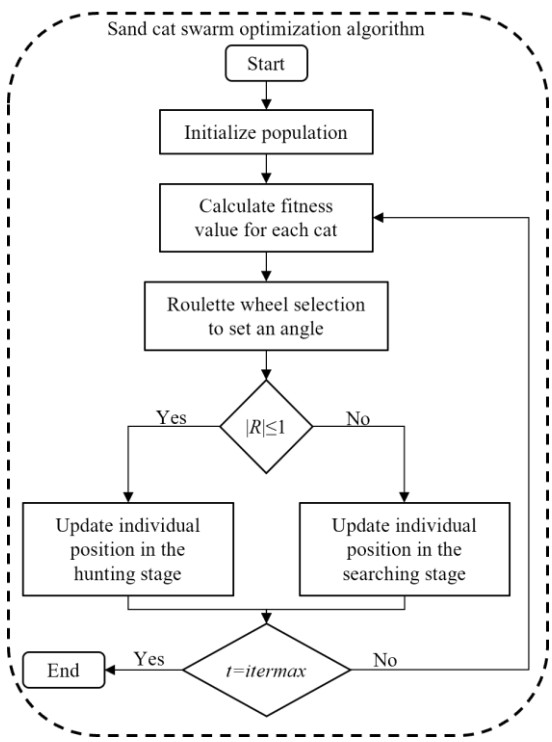

**Figure 2.** The flowchart of the sand cat swarm optimization algorithm.

### 2.3. Random Forest Algorithm

Random forest (RF) algorithm is an ensemble learning algorithm based on decision tree. The algorithm combines randomly generated decision trees to form a more stable strong classifier, whose output value is determined by the prediction results of all decision trees. Compared with the single decision tree algorithm, RF algorithm has higher prediction accuracy, stronger generalization ability, and can solve the nonlinear mapping problem between multiple parameters effectively.

### 2.3.1. Decision Tree Algorithm

Decision tree algorithm is a simple and efficient inductive learning algorithm. In regression prediction, the decision tree uses CART (classification and regression tree) algorithm [36] to judge the attribute values of internal nodes; that is, starting from the root

node, the optimal attribute is selectedaccording to the principle of minimum Gini index, and then the dichotomous recursion is used to keep building nodes down to finally form an inverted tree structure. The prediction of the decision tree is based on the path from the root node to the leaf node. Different paths of input data lead to different prediction results.

### 2.3.2. Ensemble Learning

With the rapid development of the big data era, ensemble learning has become an effective means of data mining and analysis. Its basic method is to synthesize the prediction results of several base classifiers in a certain rule or way, so as to effectively avoid the overfitting existing in a single classifier. According to whether the base classifiers are associated, ensemble learning algorithms can be divided into unrelated bagging series algorithms and associated boosting series algorithms, and RF algorithm is the representative of bagging series algorithms [37]. The classic bagging series algorithms use bootstrap to sample the original samples, train the base classifiers on several new sample sets, and finally combine all the base classifiers to obtain the final ensemble classifier. Based on traditional bagging algorithm, RF algorithm introduces random feature selection; that is, when building a base classifier (decision tree), the split attribute sets of internal nodes are randomly selected to further increase the diversity of decision tree and improve the prediction performance.

### 2.3.3. Out-of-Bag Error

The random sampling method is used to generate samples, which will inevitably lead to repeated occurrence of samples. For each new sample set, there is always a part of the original data that has not been collected during the sampling process. This part of data is called out-of-bag (OOB). Considering that out-of-bag data is not utilized in training, it can be used as the performance verification set of each decision tree. After the OOB errors of all decision trees are accumulated, the OOB error of RF can be obtained, which is able to estimate the generalization ability of the algorithm, and the OOB error has been proved to be unbiased estimates [21].

### 2.3.4. Flow of Random Forest Algorithm

1. Generate $n$ groups of training samples randomly by bootstrap method, and construct decision trees based on the new sample sets.
2. When selecting attributes for each internal node, select $m$ attributes from all attributes randomly as the attribute set of the node. Based on CART algorithm, select the optimal attribute for splitting until the decision tree grows completely. During the growth of the decision tree, pruning is not required.
3. Input the test sample set and obtain the final result on the basis of synthesizing the prediction results of all decision trees. For regression problems, the weighted average of prediction values of all decision trees is taken as the final prediction value.

The specific process of RF is shown in Figure 3.

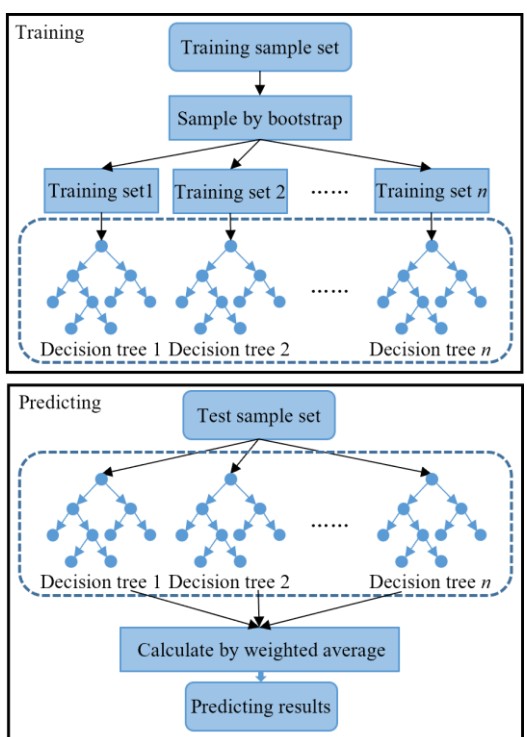

**Figure 3.** Schematic diagram of random forest algorithm.

*2.4. Construction of Deformation Prediction System of Concrete Dam Based on IVM-SCSO-RF*

2.4.1. Input Variables

According to the main causes of horizontal displacement of concrete gravity dam during operation [34], twelve system input variables are set, which are water pressure factors $(h - h_0)_u$, $(h^2 - h_0^2)_u$, $(h^3 - h_0^3)_u$, $(h - h_0)_d$, $(h^2 - h_0^2)_d$, and $(h^3 - h_0^3)_d$; temperature factors $\sin \frac{2\pi t}{365} - \sin \frac{2\pi t_0}{365}$, $\cos \frac{2\pi t}{365} - \cos \frac{2\pi t_0}{365}$, $\sin \frac{4\pi t}{365} - \sin \frac{4\pi t_0}{365}$, and $\cos \frac{4\pi t}{365} - \cos \frac{4\pi t_0}{365}$; and time effect factors $\theta - \theta_0$ and $\ln \theta - \ln \theta_0$. Here, $h$ is the water depth on the monitoring day; $h_0$ is the water depth on the initial monitoring day; the input variables consider the influence of upstream and downstream water pressure at the same time, and the subscripts $u$ and $d$ represent upstream and downstream, respectively; $t_0$ is the accumulated days from the initial modeling day to the initial monitoring day; $t$ is the accumulated days from the monitoring day to the initial monitoring day; $\theta_0$ is $0.01t_0$; $\theta$ is $0.01t$.

2.4.2. Parameters of RF

The main parameters affecting the performance of RF algorithm are the number of decision trees $n_{tree}$ and the number of variable choices to split on at each node $m_{try}$, which are simplified as $n$ and $m$, respectively. Set the OOB error of the RF algorithm as the fitness function and select $n$ and $m$ as the target parameters of the SCSO algorithm for optimization. Finally, the position corresponding to the minimum OOB error is the optimal combination of $n$ and $m$.

2.4.3. System Operation Process

1. Read the original data, establish the IVM to eliminate system interference after removing the missing items in the monitoring values, and obtain the corrected sample set.
2. Divide the sample set into training set and test set; the proportion of test set is generally 10%~20% of the total samples.
3. Input the training set data into the SCSO-RF algorithm and obtain the optimal parameter combination of the RF algorithm by SCSO algorithm.
4. Input the test set data into the RF algorithm after parameter optimization and obtain the prediction results.

5.  Analyze the prediction effect by comparing the predicted value with the actual value and calculating the sum of squared error SSE, the mean-square error MSE, the mean absolute error MAE, the root-mean-square error RMSE, and the coefficient of determination $R^2$.

To sum up, the system operation process is shown in Figure 4.

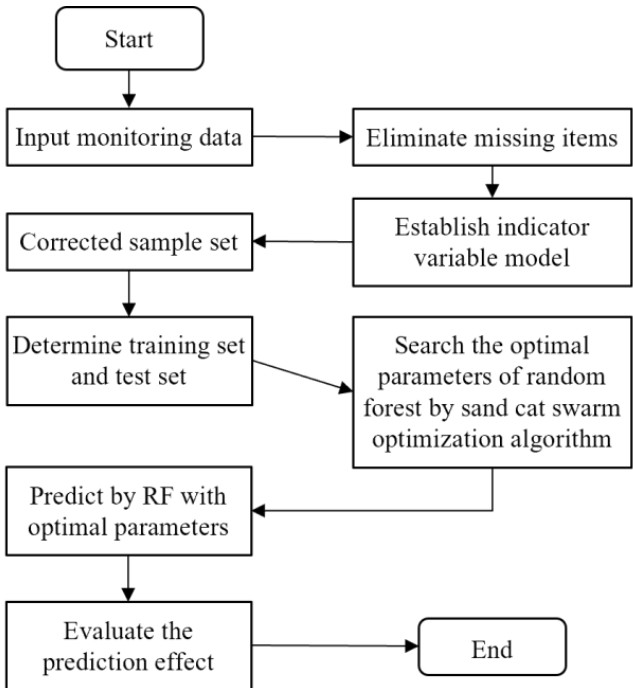

**Figure 4.** Operation process of concrete dam deformation prediction system based on IVM-SCSO-RF.

## 3. Case Study

### 3.1. Project Overview

In this study, a concrete gravity dam located in Fujian Province of China was selected as the research object. The maximum height of the dam is 101 m and the normal pool level is 65 m. The monitoring items of the dam are set up completely, the measuring points are arranged reasonably, and the working states of all monitoring facilities are generally normal. The main body of the dam is divided into 42 dam sections. Four tension lines are arranged on the dam crest. Basically, one measuring point is set for each dam section to monitor the horizontal displacement of the dam crest. Five inverted vertical lines and three vertical lines are arranged on the left and right banks and dam sections 8#, 23#, and 32# to calibrate the displacement of the end point of the tension lines, as well as to measure the deflection of typical dam sections. There are 40 measuring points of tension lines at the dam crest, numbered EX0~EX29, EX31~EX39, and EX39-1, and there are 11 measuring points of inverted vertical lines and vertical lines, numbered IPL-1, IPR-1, IP8-1, PL8-1, IP23-1~2, PL23-1~2, IP32-1~2, and PL32-1. The specific layout of measuring points is shown in Figure 5.

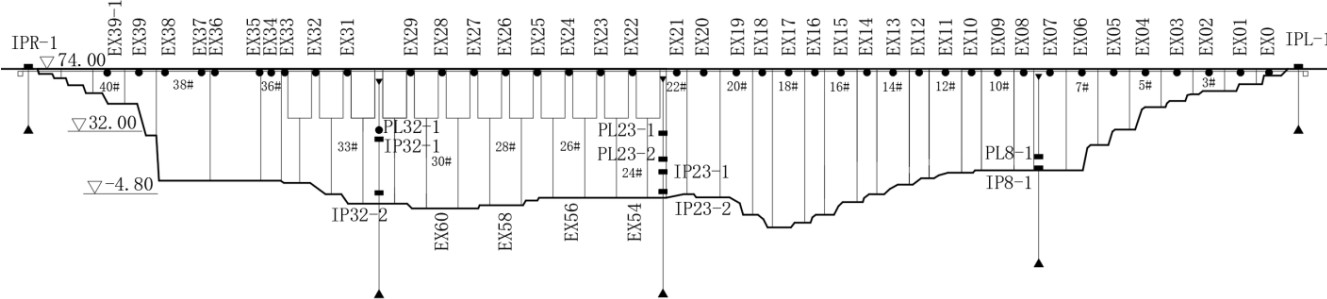

**Figure 5.** Layout of measuring points.

MATLAB® (R2020a) software was employed to implement all algorithms and models in this study. Furthermore, the statistics and machine learning toolbox and deep learning toolbox in the software were used to apply all the machine learning algorithms.

### 3.2. Performance Verification of IVM

To verify the IVM's performance in eliminating interference from the monitoring system, jump points were manually added to the normal monitoring values, and the IVM was used to correct the "interfered" monitoring values. A total of 530 groups of horizontal displacement monitoring data of measuring point PL23-1 (dam section 23#) were selected, and 3.00 mm, −6.00 mm, and 5.00 mm drift values were added, respectively, at the 140th, 270th, and 450th measured values. The IVM was established by using Equations (2)–(4), and the drift value of each jump point was calculated. The results are shown in Table 1.

**Table 1.** Calculation results of drift values.

| Change Point | Actual Drift Value (mm) | Calculated Drift Value (mm) | Absolute Error (mm) | Error Rate (%) |
|---|---|---|---|---|
| 140 | 3.00 | 3.078 | 0.078 | 2.6% |
| 270 | −6.00 | −6.037 | 0.037 | 0.6% |
| 450 | 5.00 | 4.926 | 0.074 | 1.5% |

It can be seen that the drift values calculated by IVM were very close to the manual setting values, and the maximum absolute error and error rate were only 0.078 mm and 2.6%, meeting the actual requirements. It can also be seen from Figure 6 that the correction accuracy of IVM is very high.

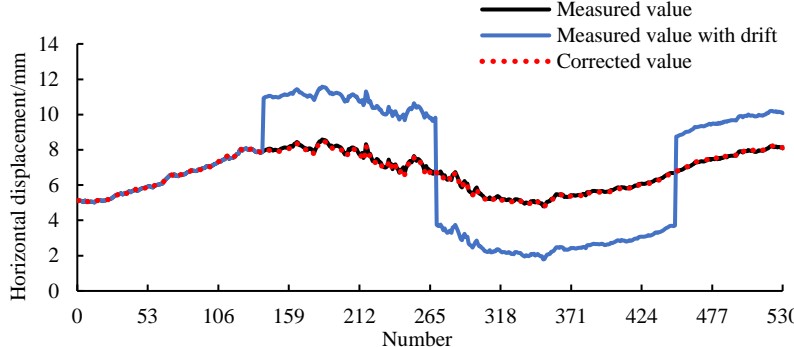

**Figure 6.** Process diagram of measured values, measured values with drift, and corrected values of PL23-1.

### 3.3. Application of Deformation Prediction System of Concrete Dam Based on IVM-SCSO-RF

To verify the performance and reliability of the deformation prediction system proposed in this paper, 660 groups of horizontal displacement monitoring data of measuring point EX19 (dam section 20#) were selected for analysis. The time range was from

12 March 2016 to 2 January 2018. The processes of upstream and downstream water levels and horizontal displacement are shown in Figures 7–9, respectively. Due to the maintenance and updating of the monitoring system, the measuring point EX19 was significantly disturbed on 26 May 2017, 14 September 2017, and 3 November 2017. The large drifts of the monitoring values at the corresponding time are marked by black circles in Figure 9. Therefore, IVM was firstly established to identify and correct abnormal monitoring data, and then the corrected monitoring data were input into the SCSO-RF model. The SCSO algorithm was used to find the optimal parameters of RF, so as to improve the prediction accuracy of the system as much as possible.

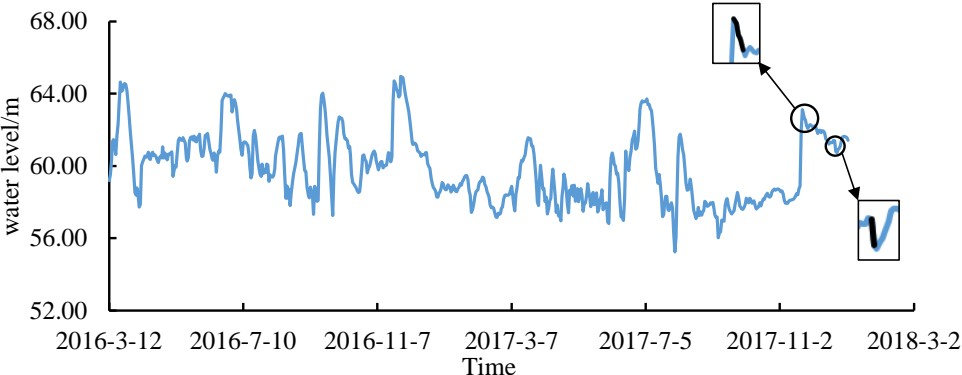

**Figure 7.** Process diagram of upstream water level.

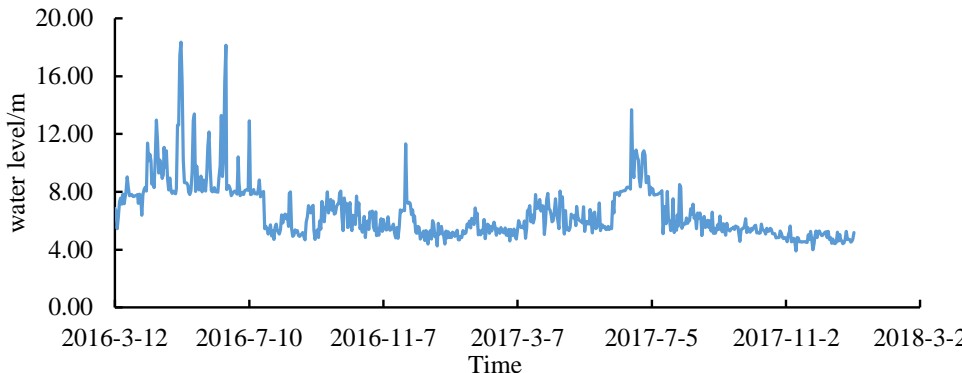

**Figure 8.** Process diagram of downstream water level.

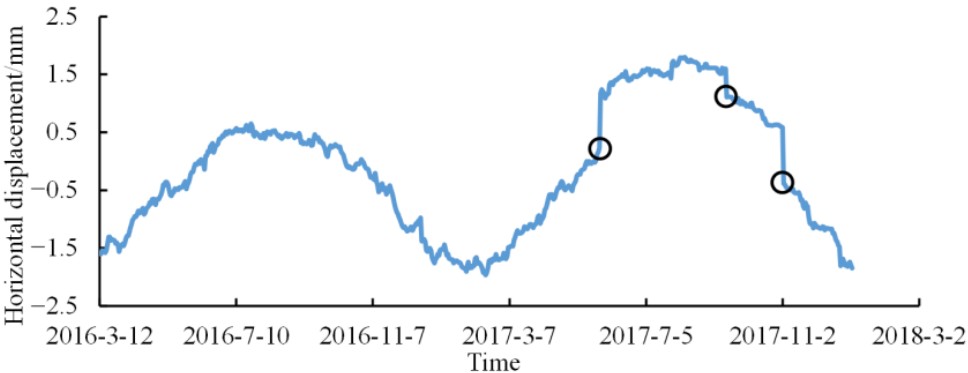

**Figure 9.** Process diagram of horizontal displacement of the measuring point EX19.

### 3.3.1. Correction of Measured Value by IVM

IVM was applied to correct the actual drift of the measuring point EX19. The process of the corrected horizontal displacement is shown in Figure 10. It can be seen that the law of the corrected monitoring value is reasonable and can better reflect the true situation of the horizontal displacement change. On the basis of the corrected horizontal displacement,

the first 528 groups of data (from 12 March 2016 to 21 August 2017) were taken as the training set for parameter optimization of RF and model training, and the last 132 groups of data (from 22 August 2017 to 2 January 2018) were taken as the test set for prediction and performance analysis.

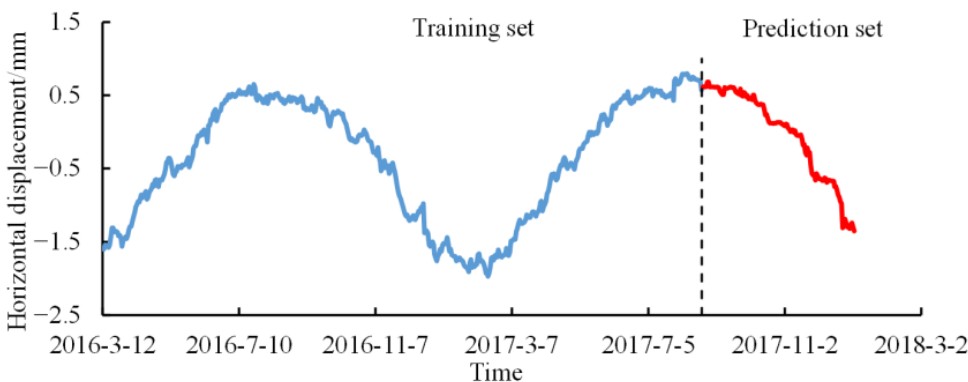

**Figure 10.** Process diagram of corrected horizontal displacement of the measuring point EX19.

### 3.3.2. Parameter Optimization of RF Algorithm

At present, the trial-and-error (TAE) method is a common method to find the optimal parameters of RF. The proposed deformation prediction system used the SCSO algorithm to optimize the parameters of RF. To verify the superiority of the SCSO algorithm in optimization performance, it was compared with TAE and the PSO algorithm.

1. The trial-and-error method

The controlling variable method is adopted generally in TAE. By default, the parameter $m$ is one-third of the total number of attributes and rounded down, and then the optimal number of decision trees $n$ is found within a given range. The total attribute number of the model in this paper was 12, so the parameter $m$ was 4. On this basis, the range of parameter n was set to [100, 1000], in which took a value every 100, and the OOB error under each parameter combination was calculated. The results are shown in Figure 11. Based on the criterion of minimum out-of-bag error, the optimization results of the trial-and-error method were $n = 400$, $m = 4$, and the minimum OOB error was $3.391 \times 10^{-3}$.

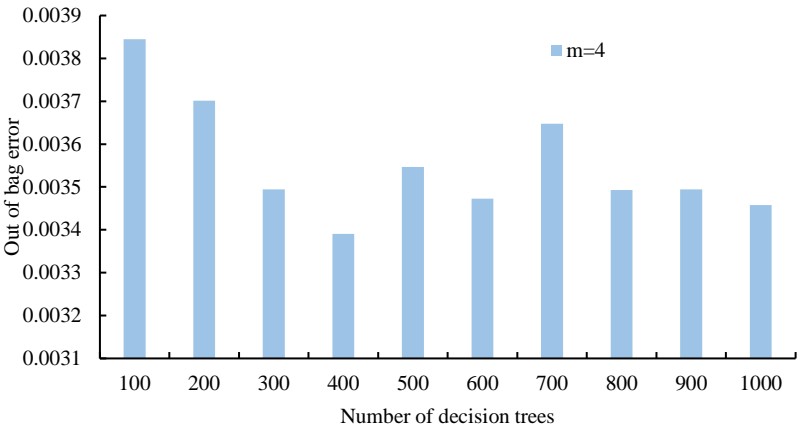

**Figure 11.** Relationship between number of decision trees and out-of-bag error in TAE.

2. Particle swarm optimization algorithm

Particle swarm optimization (PSO) algorithm is a common swarm intelligence algorithm for solving optimization problems with simple principle and wide adaptability. The maximum number of iterations of the PSO algorithm was set to 200, the total number of particles was set to 30, and the fitness function was set to be the OOB error of RF. Before optimization, the ranges of parameters $n$ and $m$ were set as [100, 1000] and [1, 12], respectively.

When the number of iterations reached the set value, the calculation was terminated. The specific iterative process of PSO-RF is shown in Figure 12. The PSO algorithm converged in the 83rd iteration, and the minimum OOB error was $2.675 \times 10^{-3}$, and the corresponding optimal parameter combination was $n = 712$, $m = 8$.

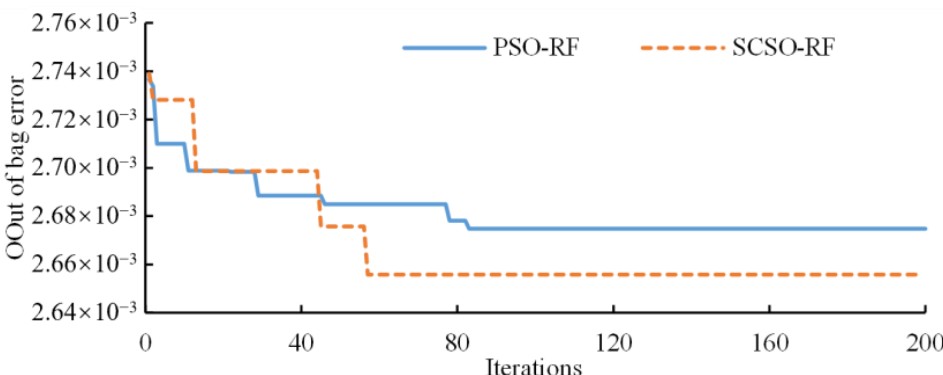

**Figure 12.** Iterative process of PSO-RF and SCSO-RF.

3.   Sand cat swarm optimization algorithm

The relevant parameters of the SCSO algorithm were set as follows: the maximum number of iterations was 200, the number of sand cats was 30, and the maximum sensitivity range was 2. Similarly, the ranges of parameters $n$ and $m$ were set as [100, 1000] and [1, 12], respectively, and the fitness function was the OOB error of RF. The specific iterative process of SCSO-RF is shown in Figure 12. It can be seen from the figure that compared with the PSO algorithm, the SCSO algorithm effectively avoided falling into the local extremum due to the introduction of linear decreasing sensitivity, random angle, random initialization and other elements, and the search accuracy and convergence efficiency significantly improved. At the 57th iteration, SCSO converged to the optimal result, the minimum OOB error was $2.656 \times 10^{-3}$, and the corresponding optimal parameter combination was $n = 437$, $m = 12$.

### 3.3.3. Training and Prediction of Model

Based on the optimal parameters obtained by three different methods, the TAE, PSO and SCSO, and RF models were constructed, respectively. At the same time, dam deformation prediction models based on long short-term memory (LSTM) and support vector machine (SVM) were set as the comparison object because they are widely used in the field of deformation prediction with high accuracy. The fitting accuracy of five models in the training set is shown in Table 2. It can be seen from the table that each model has achieved good training effect, and the fitting errors of models are small.

**Table 2.** Fitting accuracy of five models in the training set.

| Evaluation Criterion | TAE-RF | PSO-RF | SCSO-RF | LSTM | SVM |
|---|---|---|---|---|---|
| SSE/mm$^2$ | 0.742 | 0.552 | 1.209 | 3.728 | 2.136 |
| MSE/mm$^2$ | 0.001 | 0.001 | 0.002 | 0.007 | 0.004 |
| MAE/mm | 0.027 | 0.024 | 0.036 | 0.066 | 0.046 |
| RMSE/mm | 0.037 | 0.032 | 0.048 | 0.084 | 0.064 |
| R$^2$ | 0.998 | 0.998 | 0.997 | 0.990 | 0.994 |

The prediction curves of models are shown in Figure 13. It can be seen from the figure that compared with TAE-RF and PSO-RF, the predicted value of SCSO-RF has the smallest deviation from the measured value, and its variation trend is basically consistent, indicating that the SCSO algorithm has higher accuracy in the parameter optimization of RF, and the RF algorithm based on the optimal parameters has higher prediction accuracy in the dam deformation prediction. In addition, due to the advantages of ensemble learning, the

SCSO-RF model has a strong generalization ability, effectively avoiding the problem of overfitting, and its prediction performance is significantly better than the commonly used models LSTM and SVM.

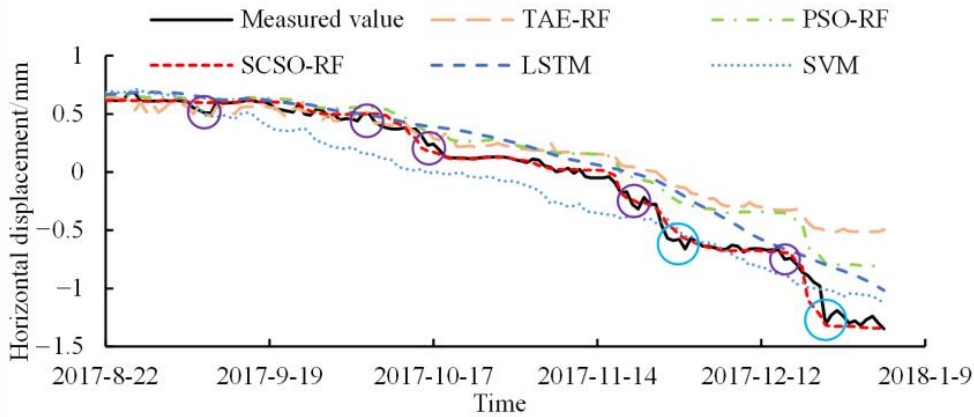

**Figure 13.** Prediction curves of models.

The residuals of each model are shown in Figure 14. It can be seen from the figure that compared with other models, the SCSO-RF model has the smallest deviation between the residuals and the zero value, and it has obvious advantages in prediction accuracy. Furthermore, the distribution of residuals of the SCSO-RF model is more concentrated, and its fluctuation range is the smallest, indicating that the model has strong stability.

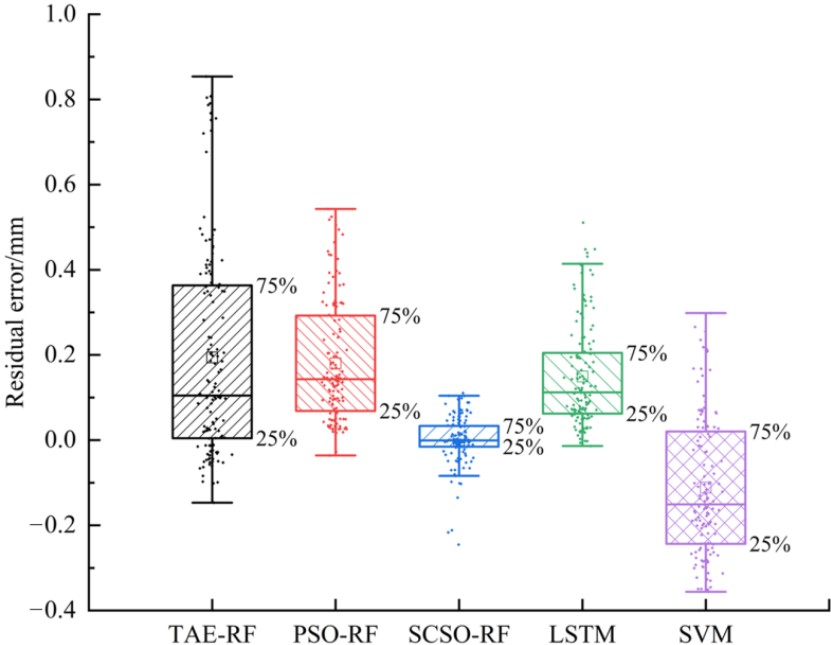

**Figure 14.** Comparison of residuals of models.

The results of the evaluation criteria of each model are shown in Table 3. It can be seen from the table that the prediction accuracy of the TAE-RF is the lowest in this case study, with its SSE and MAE of 12.810 mm$^2$ and 0.217 mm, respectively. It lost a lot of characteristic information of the monitoring data because this method selected parameters of RF relying on subjective experience, and thus the stability of the model is poor, with its MSE and RMSE of 0.097 mm$^2$ and 0.312 mm, respectively, and R$^2$ of 0.731. The prediction accuracy and stability of RF optimized by PSO algorithm have been improved. Compared with TAE-RF, the SSE, MSE, MAE, and RMSE of PSO-RF are reduced by 47%, 46%, 17% and 27%, respectively, and R$^2$ is 0.856. However, due to falling into the local extreme value

in parameter optimization, the RF optimized by the PSO algorithm does not give full play to its best performance. The common LSTM and SVM are similar in this case, and the prediction accuracy and stability of LSTM are slightly better than those of SVM. Compared with PSO-RF, the SSE, MSE, MAE, and RMSE of LSTM are reduced by 29%, 29%, 17%, and 16% respectively, and its $R^2$ is close to 0.9, but the prediction accuracy is still not satisfactory. With powerful capability of global optimization, the SCSO algorithm has improved the prediction performance of RF greatly. The SSE, MSE, MAE, and RMSE of SCSO-RF have all reached the minimum of all models. The four evaluation criteria have decreased by 91%, 92%, 76%, and 71% respectively, compared with the LSTM with better prediction accuracy, and by 97%, 97%, 83%, and 82%, respectively, compared with the TAE-RF with the worst prediction accuracy. Moreover, the $R^2$ of SCSO-RF is very close to 1. The results further verify the reliability and prediction performance of the deformation prediction system proposed in this paper.

**Table 3.** Comparison of prediction performance of models.

| Evaluation Criterion | TAE-RF | PSO-RF | SCSO-RF | LSTM | SVM |
|---|---|---|---|---|---|
| SSE/mm$^2$ | 12.810 | 6.851 | 0.418 | 4.857 | 5.016 |
| MSE/mm$^2$ | 0.097 | 0.052 | 0.003 | 0.037 | 0.038 |
| MAE/mm | 0.217 | 0.181 | 0.036 | 0.150 | 0.170 |
| RMSE/mm | 0.312 | 0.228 | 0.056 | 0.192 | 0.195 |
| $R^2$ | 0.731 | 0.856 | 0.991 | 0.898 | 0.895 |

To better understand the capability of the model in this paper, 12 time points with drastic changes in the measured displacement are selected to evaluate models. The specific positions of these points are marked by blue and purple circles in Figure 13, and the prediction results of models at the specified points are shown in Table 4. It can be seen from Table 4 that the prediction results of SCSO-RF at 10 designated points are the closest to the measured values compared with other models. Although its prediction results on 9 September 2017 and 25 November 2017 are not as good as those of the SVM, the maximum deviation is only 0.10 mm, which also achieves high prediction accuracy. The results of the evaluation criteria of models at the specified points are shown in Table 5. The prediction performance of each model is further analyzed by comparing the evaluation criteria at these points with large changes of displacement. It can be seen from Table 5 that the difference in prediction performance of each model is further amplified at these points. The prediction accuracy and stability of SCSO-RF at the specified points are significantly better than those of all other models. Compared with the TAE-RF with the worst prediction performance, the SSE, MSE, MAE, and RMSE of SCSO-RF are reduced by 97%, 97%, 83%, and 83%, respectively, and $R^2$ is still very close to 1, which indicates that the model can mine the effective information in the input variables fully and make more accurate prediction for large changes in displacement caused by variations of environmental factors. Furthermore, the performance of LSTM at these points is significantly worse than that of the global prediction, indicating that the model ignores part of the effective information in the input variables.

In addition, the anomalous behavior of displacement at the end of November and at the end of December 2017 (marked by blue circles in Figure 13) needs to be analyzed because anomalies may be related to precipitation data, strong winds causing so-called seiches, geological displacements along existing faults, damage/microcracking due to aging, etc. [38,39], the analysis of which can give useful information on the physical/geological agents affecting dam stability and on the state of the dam. According to the results of field investigation and monitoring data, the dam was not affected by strong wind, geological displacement, damage/cracks, and other factors, and the sudden drops of displacement that happened on 25~29 November 2017 and 23 December 2017 were mainly caused by the change in upstream water level. It can be seen from Figure 7 that the upstream water

level decreased significantly on 22~26 November 2017 and 21~22 December 2017 (marked by black lines), which led to the rapid reduction in the upstream water pressure, and thus increased the speed of change in displacement to upstream direction, namely the sudden drop of displacement. Furthermore, it can be clearly seen that the influence of water level on displacement has a hysteresis of about 2–3 days.

**Table 4.** Predicted values of models at the specified points.

| Time | Measured Value/mm | Predicted Value/mm | | | | |
|---|---|---|---|---|---|---|
| | | **TAE-RF** | **PSO-RF** | **SCSO-RF** | **LSTM** | **SVM** |
| 9 September 2017 | 0.51 | 0.48 | 0.62 | 0.60 | 0.63 | 0.48 |
| 6 October 2017 | 0.50 | 0.51 | 0.56 | 0.50 | 0.49 | 0.16 |
| 16 October 2017 | 0.23 | 0.35 | 0.35 | 0.18 | 0.39 | −0.01 |
| 20 November 2017 | −0.28 | 0.05 | −0.04 | −0.24 | −0.01 | −0.39 |
| 21 November 2017 | −0.32 | 0.03 | −0.08 | −0.26 | −0.02 | −0.38 |
| 25 November 2017 | −0.46 | −0.04 | −0.14 | −0.36 | −0.10 | −0.43 |
| 26 November 2017 | −0.57 | −0.09 | −0.21 | −0.46 | −0.12 | −0.41 |
| 27 November 2017 | −0.59 | −0.11 | −0.22 | −0.49 | −0.15 | −0.45 |
| 28 November 2017 | −0.58 | −0.13 | −0.26 | −0.54 | −0.18 | −0.51 |
| 29 November 2017 | −0.66 | −0.14 | −0.27 | −0.57 | −0.21 | −0.56 |
| 16 December 2017 | −0.75 | −0.33 | −0.35 | −0.69 | −0.66 | −0.89 |
| 23 December 2017 | −1.31 | −0.50 | −0.78 | −1.32 | −0.80 | −1.01 |

**Table 5.** Comparison of prediction performance of models at the specified points.

| Evaluation Criterion | **TAE-RF** | **PSO-RF** | **SCSO-RF** | **LSTM** | **SVM** |
|---|---|---|---|---|---|
| SSE/mm$^2$ | 2.189 | 1.204 | 0.060 | 1.359 | 0.365 |
| MSE/mm$^2$ | 0.182 | 0.100 | 0.005 | 0.113 | 0.030 |
| MAE/mm | 0.369 | 0.287 | 0.062 | 0.297 | 0.144 |
| RMSE/mm | 0.427 | 0.317 | 0.071 | 0.337 | 0.174 |
| R$^2$ | 0.307 | 0.619 | 0.981 | 0.570 | 0.884 |

## 4. Discussion

In the data preprocessing of building the deformation prediction system of concrete dams, only the interference to the observation system, caused by engineering reinforcement, maintenance, and updating of the monitoring system, which are common in practical projects, was considered. The case study showed that the indicator variable model could achieve high-precision fitting and accurate removal of the obvious drift phenomenon appearing in the measured values. However, in the actual application process, the correction accuracy of the indicator variable model is related to the observation accuracy and model input variables. The higher the observation accuracy is and the more comprehensive the input variables are, the higher the correction accuracy will be. On the contrary, the correction accuracy may be low. Therefore, before establishing the indicator variable model, it is necessary to ensure that the accuracy of the observation system meets the actual requirements as much as possible and to consider all aspects when selecting the input variables.

In addition, in the actual monitoring data, there are also many data anomalies other than drift of measured value, such as data loss, frequent fluctuations, and high noise. If these data anomalies are not identified and processed in time, they will have greater side effects on the training and prediction of the system. Although there are many studies on the preprocessing of dam monitoring data, most of them are based on unilateral processing of unusual data, which do not involve identification and correction of all data anomalies. Therefore, the establishment of an all-round and real-time identification and processing system of data anomalies is an important problem that needs to be solved urgently in the future, which is the premise and guarantee for automatic online evaluation of dam safety behavior.

Comparing the prediction results of TAE-RF, PSO-RF, and SCSO-RF, it can be seen that the selection of parameters has a great impact on the prediction accuracy of the RF algorithm. Water pressure, temperature, and time effect constitute the main causes of horizontal displacement change of concrete gravity dams during operation. Since all 12 input variables selected in this study have certain importance in deformation prediction, the fewer input variables considered in the training of RF, the worse the prediction effect of the model will theoretically be. The default of parameter $m$ in TAE was one-third of the total number of attributes and rounded down, which was $m = 4$, which meant that RF only extracted 4 variables from 12 input variables randomly as the attribute set of each internal node in the process of generating the decision tree, which would lead to the loss of a large amount of feature information in the variables, further resulting in low prediction accuracy. It can be seen that in the application process of the RF algorithm, the settings of default parameters are not necessarily reasonable and need to be carefully verified and analyzed according to the actual situation. The prediction effects of PSO and SCSO were better than TAE because they were optimized within the specified parameter range of RF. However, the PSO algorithm did not find the parameter combination with the minimum OOB error because it was trapped in local extremum, and its prediction accuracy was lower than that of the SCSO algorithm. In addition, the optimal parameter $m$ of RF obtained by SCSO was 12, which meant that all attributes were attribute sets of internal nodes. It indicates that all 12 input variables contain key information for predicting deformation, but this is only the optimization result of measuring point EX19 in this study, and it may not be applicable to other measuring points and concrete dams. To sum up, in the application process of the RF algorithm, the introduction of intelligent optimization algorithms with strong performance such as SCSO for parameter optimization is the important way to improve the prediction accuracy of the model. To further improve the optimization efficiency and shorten the optimization time will be the main research content in the future.

## 5. Conclusions

The performance of the RF algorithm in the prediction of concrete dam deformation is affected by many factors, such as drift of the measured value and the inappropriate setting of parameters of RF, which may greatly affect the prediction accuracy and stability of the RF algorithm. To solve the above problems, a deformation prediction system based on the indicator variable model and the RF algorithm optimized by SCSO is proposed in this work. The system realizes the adaptive correction of monitoring data through IVM, and uses the SCSO algorithm to search for the optimal parameters of RF. On this basis, the concrete dam displacement is predicted with convincing accuracy and stability.

The case study shows that IVM can accurately identify and eliminate the interference to the automatic monitoring system based on the actual information of the dam, and its maximum error rate is less than 3%. The corrected monitoring data can better reflect the real change of displacement. The optimal parameters of RF are searched on the basis of corrected data. The results show that the results of the SCSO algorithm are obviously better than those of the TAE method and PSO algorithm, and the corresponding OOB error is the smallest, which indicates that the SCSO algorithm has a strong capability of global optimization and is more suitable for solving complex optimization problems. A deformation prediction model of the concrete dam is built based on RF optimized by SCSO, and the performance of prediction is compared with TAE-RF, PSO-RF, LSTM, and SVM. The results show that the residual distribution of SCSO-RF is the most reasonable, and the five quantitative evaluation criteria are better than other models. Compared with other models, the SSE and MSE of SCSO-RF are reduced by at least 91%, and the MAE and RMSE are reduced by at least 71%. Moreover, the $R^2$ of SCSO-RF is very close to 1. At the same time, SCSO-RF has more obvious advantages in the prediction of designated points with large changes of displacement. The above results fully verify that the RF algorithm optimized by the SCSO algorithm has excellent abilities of nonlinear data mining and prediction, which can be widely used in practical projects. In addition, the deformation prediction system

proposed in this paper has strong applicability and can be applied to other types of concrete dams with a little modification.

There are many data anomalies in the actual monitoring data, which may have great side effects on the training and prediction of the deformation prediction system. Only the correction of drift in the monitoring data is considered in this paper, and the all-round identification and processing of data anomalies need to be further supplemented in the model. In addition, the differences between measured displacements and calculated curves of the model need to be further studied in the following work, because the analysis of these differences can give useful information on the physical/geological agents affecting dam stability and on the state of the dam.

**Author Contributions:** Conceptualization, S.Z. and D.Z.; methodology, D.Z. and S.Z.; software, S.Z.; validation, S.Z. and Y.L.; investigation, D.Z.; data curation, S.Z. and Y.L.; writing—original draft preparation, S.Z.; writing—review and editing, Y.L.; funding acquisition, D.Z. All authors have read and agreed to the published version of the manuscript.

**Funding:** This research was funded by the National Key R&D Program of China (2018YFC1508603, 2018YFC0407104, 2018YFC0407101), National Natural Science Foundation of China (Grant Nos. 52179128, 51579083, 51579085, 51739003).

**Data Availability Statement:** Not applicable.

**Acknowledgments:** The authors are grateful to the reviewers and editors for their helpful and constructive comments, which significantly improved our manuscript.

**Conflicts of Interest:** The authors declare no conflict of interest.

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
