# Peer review of "Deformation Prediction System of Concrete Dam Based on IVM-SCSO-RF"

_water, doi:10.3390/w14223739_

Round 1
Reviewer 1 Report
This manuscript, water-2030978-peer-review-v1- entitled "Deformation Prediction System of Concrete Dam Based on IVM‐SCSO‐RF," is well written and has potential, but it should be more organized. This research investigates the concrete dam deformation prediction system based on an indicator variable model (IVM) and random forest (RF) algorithm optimized by cat swarm optimization (SCSO).
In my opinion, a careful revision of the English language should be carried out as there currently are some unclear sentences. The study seems to be well-designed. The methodology and results are technically sound. Discussions on the scientific and practical values of the study, the limitations of proposed models, and future work are meaningful. I recommend accepting this manuscript after revision. The main concerns are as follows:
1) Quantitative results should be provided in the abstract to make it more comprehensive. The results of the models should be added in the abstract section. Also, The main aim of the study should be clearly mentioned in the abstract.
2) More literature review about the other methods is needed. The manuscript could be substantially improved by relying and citing more on recent literature about contemporary real-life case studies of sustainability and/or uncertainty, such as the followings.
· Samani, S., Vadiati, M., Azizi, F., Zamani, E., & Kisi, O. (2022). Groundwater Level Simulation Using Soft Computing Methods with Emphasis on Major Meteorological Components. Water Resources Management, 36(10), 3627-3647.
· Vadiati, M., Rajabi Yami, Z., Eskandari, E., Nakhaei, M., & Kisi, O. (2022). Application of artificial intelligence models for prediction of groundwater level fluctuations: Case study (Tehran-Karaj alluvial aquifer). Environmental Monitoring and Assessment, 194(9), 1-21.
3) For readers to quickly catch your contribution, it would be better to highlight significant difficulties and challenges and your original achievements to overcome them more straightforwardly in the abstract and introduction.
4) Providing a comprehensive flowchart is highly recommended by researchers, so please add a flowchart representing the methodology in the paper. Fig. 2 can be improved and extended to address all used models in this study.
5) What are other feasible alternatives for the case study? What are the advantages of adopting this case study over others in this case? How will this affect the results? The authors should provide more details on this.
6) Please provide all software used in this study. Which library or pachage did you use to apply the methods?
7) Tab. 3 is the most important table in the manuscript, and, unfortunately, the authors did not try to discuss it in a specific way. A comprehensive discussion emphasizing would significantly improve the paper on the table.
8) It is better to add more error criteria such to better understand the model's ability.
9) It seems that conclusions are observations only, and the manuscript needs thorough checking for explanations given for results. The authors should interpret more precisely the results argument.
Reviewer 2 Report
The preprocessing of data should be done very carefully as the important details can be eliminated as the result of All data anomalies should be documented and analyzed properly - namely, compared with engineering inferences, dam damage due to aging/cracking - especially important for old dams (see Guyer R, Johnson P. Nonlinear Mesoscopic Elasticity. 2009. Wiley-VCH), anomalous weather, geological displacement on the nearby fault, seismic events, etc.
It is interesting to calculate the difference between measured displacements and SCSO-RF calculated curves and compare anomalies with precipitation data, strong winds, causing so-called seiches, geological displacements along existing faults, damage/micro-cracking due to aging (see Guyer R, Johnson P. Nonlinear Mesoscopic Elasticity. 2009. Wiley-VCH), dam engineering inferences - say, WL regulations, etc. (see, for example, T. CHELIDZE, T. MATCHARASHVILI, V. ABASHIDZE, M. KALABEGISHVILI, N. ZHUKOVA. 2013. Real time monitoring for analysis of dam stability: Potential of nonlinear elasticity and nonlinear dynamics approaches. Front. Struct. Civ. Eng. DOI 10.1007/s11709-013-0199-5). The analysis of reactions to these impacts can give useful information on the physical/geological agents affecting dam stability and on the state of dam. By the way, the dam is not an isolated structure – its foundation is connected to some geological formation, which has its own displacement manner, which affects also dam structure.
Lastly, I'd like to comment a bit the results of prediction curves Fig. 11 - the performance of SCSO-RF model is outstanding! At the same time, in order to make not just the perfect mathematical fitting, it is desirable to find the physical reasons of anomalous deflections in the Fig.11 - say, to discuss the anomalous behavior of displacement at the end of November and at the end of December 2017 - may be this is the effect of strong rain or of some other actors – see above.
The above remarks are rather the suggestions for the following investigations: I fully agree with the remark of authors: “Therefore, the establishment of an all‐round and real‐time identification and processing system of data anomalies is an important problem that needs to be solved urgently in the future, which is the premise and guarantee for automatic online evaluation dam safety behavior”.
The paper can be published as it is presented, but I recommend also to mention of some related papers, published in leading journals:
Influence of periodic variations in water level on regional seismic activity around a large reservoir: Field data and laboratory model. J. Peinke et al. (2006) Physics of the Earth and Planetary Interiors 156 130–142
Investigating the dynamical features of the time distribution of the reservoir-induced seismicity in Enguri area (Georgia). L. Telesca et al. Nat Hazards DOI 10.1007/s11069-013-0855-z
Complex dynamics of fault zone deformation under large dam at various time scales. T. Chelidze et al. 2019. Geomech. Geophys. Geo-energ. Geo-resour. (GGGG). https://doi.org/10.1007/s40948-019-00122-3
Time Series Analysis of Fault Strain Accumulation Around Large Dam: The Case of Enguri Dam, Greater Caucasus. T. Chelidze et al. 2020. In “Building Knowledge for Geohazard Assessment and Management in theCaucasus and other Orogenic Regions” Ed. F. Bonali et al. Springer Nature. Pp 31-40.
Potential of Nonlinear Dynamics Tools in the Real Time Monitoring of Large Dams: the case of High Enguri Arc Dam. T. Chelidze et al. Volume “Dam Engineering” Ed. H. Tusun. Intech Open. 2022.
Round 2
Reviewer 1 Report
The authors addressed my comments.
Reviewer 2 Report
I think author's corrections are good